# ProFertil study protocol for the investigation of gonadotropin-releasing hormone agonists (GnRHa) during chemotherapy aiming at fertility protection of young women and teenagers with cancer in Sweden—a phase III randomised double-blinded placebo-controlled study

Kenny A Rodriguez-Wallberg [1,2] Hanna Pauline Nilsson [1] Jonas Bergh,[1,3] Johan Malmros,[4] Per Ljungman,[5,6] Theodoros Foukakis,[1,3] Christina Linder Stragliotto,[3] Erika Isaksson Friman,[7] Barbro Linderholm,[8] Antonis Valachis [9] Anne Andersson [10] Sara Harrysson,[11] Lovisa Vennström,[12] Per Frisk,[13] Helena Mörse,[14] Sandra Eloranta[15,16]

For numbered affiliations see end of article.

**Correspondence to**
Kenny A Rodriguez-Wallberg;
kenny.rodriguez-wallberg@ki.se

## ABSTRACT

**Background** Gonadotropin-releasing hormone agonists (GnRHa) cotreatment used to transiently suppress ovarian function during chemotherapy to prevent ovarian damage and preserve female fertility is used globally but efficacy is debated. Most clinical studies investigating a beneficial effect of GnRHa cotreatment on ovarian function have been small, retrospective and uncontrolled. Unblinded randomised studies on women with breast cancer have suggested a beneficial effect, but results are mixed with lack of evidence of improvement in markers of ovarian reserve. Unblinded randomised studies of women with lymphoma have not shown any benefit regarding fertility markers after long-term follow-up and no placebo-controlled study has been conducted so far. The aim of this study is to investigate if administration of GnRHa during cancer treatment can preserve fertility in young female cancer patients in a double-blind, placebo-controlled clinical trial.

**Methods and analysis** A prospective, randomised, double-blinded, placebo-controlled, phase III study including 300 subjects with breast cancer. In addition, 200 subjects with lymphoma, acute leukemias and sarcomas will be recruited. Women aged 14–42 will be randomised 1:1 to treatment with GnRHa (triptorelin) or placebo for the duration of their gonadotoxic chemotherapy. Follow-up until 5 years from end of treatment (EoT). The primary endpoint will be change in anti-Müllerian hormone (AMH) recovery at follow-up 12 months after EoT, relative to AMH levels at EoT, comparing the GnRHa group and the placebo group in women with breast cancer.

**Ethics and dissemination** This study is designed in accordance with the principles of Good Clinical Practice

## STRENGTHS AND LIMITATIONS OF THIS STUDY

⇒ Double-blinded randomised placebo-controlled study.
⇒ The study will recruit both teenagers and young women to cover the full span of female fertility.
⇒ The study will include several common cancer diagnoses among young females.
⇒ The study is underpowered to detect diagnosis-specific differences outside the subcohort with breast cancer.
⇒ The study has a long follow-up and might suffer from higher than estimated drop-out rates.

(ICH-GCP E6 (R2)), local regulations (ie, European Directive 2001/20/EC) and the ethical principles of the Declaration of Helsinki. Within 6 months of study completion, the results will be analysed and the study results shall be reported in the EudraCT database.

**Study registration** The National Institutional review board in Sweden dnr:2021–03379, approval date 12 October 2021 (approved amendments 12 June 2022, dnr:2022-02924-02 and 13 December 2022, dnr:2022-05565-02). The Swedish Medical Product Agency 19 January 2022, Dnr:5.1-2021-98927 (approved amendment 4 February 2022). Manufacturing authorisation for authorised medicinal products approved 6 December 2021, Dnr:6.2.1-2020-079580. Stockholm Medical Biobank approved 22 June 2022, RBC dnr:202 253.

**Trial registration number** NCT05328258; EudraCT number:2020-004780-71.

## INTRODUCTION
### Ovarian protection

The utilisation of gonadotropin-releasing hormone agonists (GnRHa) as a means to protect the ovaries from chemotherapy[1] is becoming increasingly common worldwide, even in subjects undergoing less gonadotoxic chemotherapeutic treatments, such as those indicated for benign diseases. But the effect of a GnRHa co-treatment and the mechanism for transient suppression of the ovarian function during chemotherapy are still under debate.[2–5]

The primary mechanism proposed for GnRHa-mediated protection of the ovarian reserve involves keeping the primordial follicles dormant throughout chemotherapy by antagonistic occupation of the GnRH receptor, leading to desensitisation of the pituitary gland and suppression of follicle stimulating hormone (FSH) and luteinising hormone (LH) secretion, with a profound hypogonadal effect achieved through receptor downregulation by internalisation of receptors. But primordial follicles are non-proliferating, and not regulated by gonadotropins as they lack both FSH and GnRH receptors.[6] Thus, administration of GnRHa should not be able to block their initial recruitment towards maturation, at least not through the proposed mechanism.[2–5] If GnRHa has an antagonistic effect on the ovarian cytotoxicity of chemotherapeutics, as has been suggested, systemic use of GnRHa would likely reduce the overall effect of chemotherapy, which does not seem to be the case.[7 8] Nevertheless, it is possible that prophylactic GnRHa therapy acts through mechanisms yet unknown. A recent scientific debate article has highlighted the inconsistencies among research publications,[5] and the American Society of Clinical Oncology (ASCO)'s expert panel did not recommend ovarian suppression using GnRHa as a method for fertility preservation in its updated guidelines for fertility preservation for patients with cancer published in July 2018.[7] However, several medical societies, such as the European Society for Medical

**Table 1** Data on biochemical markers of ovarian reserve from randomised controlled studies investigating GnRHa cotreatment during gonadotoxic treatment in women with cancer

| Type of cancer | Author, year | Ovarian marker studied | Protective effect (yes/no) | Study results |
|---|---|---|---|---|
| Hodgkin lymphoma | Nitzschke et al, 2010[22] | AMH and inhibin | No | Follow-up 2.7 years. No difference between the groups. |
| Hodgkin lymphoma | Behringer et al, 2010[17] | AMH and inhibin | No | Follow-up 18 months. Ovarian markers reduced in both arms. |
| Breast cancer | Munster et al, 2010[21] | Inhibin | No | Follow-up 18 months. Two pregnancies in the control group. |
| Breast cancer | Gerber et al, 2011[20] | AMH and inhibin | No | Follow-up 6 months. Ovarian markers reduced in both arms. |
| Breast cancer | Elgindy et al, 2011[19] | AMH | No | Follow-up 12 months. Ovarian marker reduced in both groups. |
| Breast cancer | Del Mastro et al, 2011[10] | No sensitive ovarian markers investigated. FSH randomly assessed | Yes | Follow up 12 months. Menstruation resumption reported but subjectively in an unblinded study. |
| Breast cancer | Moore et al, 2015 and 2019 [11 12] | No sensitive ovarian markers investigated, FSH and inhibin measured but results not reported | Yes | Study originally powered for 416 patients. Number reported by ITT 218 with only 135 with complete data after 2 years follow up. Menstruation resumption reported but in an unblinded study. Women attempted and achieved pregnancy to a higher degree in the group that received GnRHa. |
| Breast cancer | Leonard et al, 2017[16] | AMH and FSH | Yes | Follow up 24 months. Significant reduction of amenorrhoea. No significant difference in AMH. |
| Lymphoma | *Demeestere et al, 2016*[18] | AMH and inhibin | No | Follow-up 7 years. No effect on AMH or in pregnancies. |

AMH, anti-Müllerian hormone; FSH, follicle stimulating hormone; GnRHa, gonadotropin-releasing hormone agonists; ITT, intention-to-treat.

**Table 2** Primary and secondary objectives

| Primary objective | |
|---|---|
| | To estimate the changes in ovarian reserve following chemotherapy for treatment of cancer with or without GnRHa by determination of the anti-Müllerian hormone (AMH) at 12 months after end of gonadotoxic treatment (EoT) in women with breast cancer. |
| **Secondary objectives** | |
| Key secondary objective | To estimate the changes in ovarian reserve following chemotherapy for treatment of cancer with or without GnRHa by determination of the AMH at 12 months after EoT in women with acute leukemias, lymphomas and sarcomas. |
| Secondary objectives (to be evaluated at EoT, 6, 12 months after EoT and continuously during follow-up years 2, 3, 4, 5 after EoT) | To estimate the changes in ovarian reserve with or without GnRHa by determination of the antral follicle counts (AFC) in women with breast cancer. |
| | To estimate the changes in ovarian reserve with or without GnRHa by determination of the AFC in women with acute leukemias, lymphomas and sarcomas. |
| | To estimate the changes in ovarian reserve with or without GnRHa by longitudinal observation of AMH levels in women with breast cancer. |
| | To estimate the changes in ovarian reserve with or without GnRHa by longitudinal observation of AMH levels in women with acute leukemias, lymphomas and sarcomas. |
| | To compare the proportion of females with or without GnRHa that develops ovarian insufficiency by determination of follicle stimulating hormone (FSH), inhibin and estradiol at standardised timepoints. |
| | To investigate the impact of body mass index (BMI), use of contraceptives and endocrine adjuvant therapy in changes of ovarian reserve with or without GnRHa by longitudinal observation of AMH levels, FSH, inhibin and estradiol at standardised timepoints. |
| | To investigate the effect of GnRHa on ovarian blood supply. |
| | To estimate the proportion of females with or without GnRHa that develop amenorrhoea (no menstruations). |
| | To investigate fertility and childbirth after cancer treatment in women with or without GnRHa who attempt pregnancy during follow-up. |
| | To determine health-related quality of life (QoL) with or without GnRHa during chemotherapy and after cancer treatment. |
| | To study development of comorbidities during follow-up and bone mineral density after completion of cancer treatment with or without GnRHa. |
| | To estimate disease-specific oncologic outcomes, recurrence rate, overall survival and disease-free survival with or without GnRHa. |

GnRHa, gonadotropin-releasing hormone agonists.

Oncology (ESMO)[8] are currently advocating its utilisation, alone or in combination with fertility-preservation strategies for breast cancer patients treated with (neo) adjuvant chemotherapy.[8] In many countries, the GnRHa treatment is offered because of its relatively low cost in comparison to the higher expenses surrounding proven methods for fertility preservation, such as gonadotropin stimulation treatment aimed at cryopreservation of eggs or embryos or the surgical retrieval of ovarian tissue for cryopreservation.[7]

### Clinical studies on the protective effect of GnRHa
Most clinical studies supporting a beneficial effect of GnRHa cotreatment on ovarian function have been small, mainly retrospective and uncontrolled, and have used resumption of menses as surrogate to ovarian function after chemotherapy.[5] There are randomised studies of GnRHa cotreatment during chemotherapy in women with cancer, most of small size.[9–22] Importantly, as all available studies have been unblinded, the women who received GnRHa have known that the drug was administered to protect their fertility potential. This potential bias has been discussed in the literature, as the women who received GnRHa also reported higher frequency of pregnancy attempts.[2 11]

Serum AMH concentration is currently considered the gold standard marker of ovarian reserve[23–25] and since 2010 it has been included as an outcome measure in some of the randomised studies available on GnRHa cotreatment.[16–22] None of the studies investigating changes in ovarian markers following chemotherapy have found a beneficial effect of GnRHa cotreatment on AMH recovery.[16–22] Results from available randomised controlled studies that assessed markers of ovarian reserve are shown in table 1.

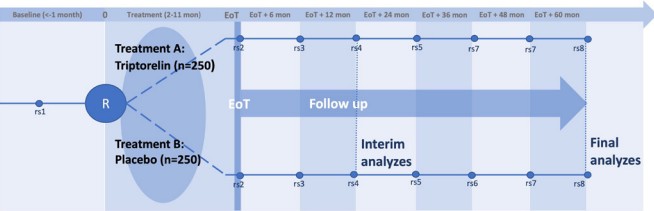

**Figure 1** Study time line. Baseline, screening/enrolment; R, randomisation, start of study treatment (–14 to +7 days from start of gonadotoxic treatment); EoT, 2–11 months after initiation of gonadotoxic treatment depending on disease-specific protocol; rs, research sample (including blood sample to measure ovarian markers), transvaginal ultrasound and QoL. EoT, end of treatment; QoL, quality of life.

## Study rationale

Among all sequelae of cancer treatment, infertility is recognised to have a high negative impact on quality of life in cancer survivors. If GnRHa treatment can be associated with maintained fertility, it would be particularly suitable for the young subjects, or for subjects that require immediate cancer therapy and do not have the opportunity to perform established and time-demanding fertility preservation procedures. However, if the GnRHa treatment does not show any protective effects, available resources should be devoted to established methods of fertility preservation and research should focus on finding new therapies that can prevent infertility in our youngest cancer patients.

The bulk of data on GnRHa cotreatment in women receiving chemotherapy for treatment of cancer is originated from two specific patient groups, those of women with breast cancer and lymphoma (Hodgkin's lymphoma and non-Hodgkin's lymphoma), as summarised in table 1. We aim to estimate the changes in ovarian reserve following chemotherapy for treatment of cancer with or without GnRHa by determination of the AMH at 12 months after end of gonadotoxic treatment (EoT) in women with breast cancer, but the study will also include subjects receiving less gonadotoxic treatment, such as the primary treatment of acute leukaemia, as well as subjects receiving very high gonadotoxic treatment, such as treatment of sarcoma, both diseases with a high incidence among teenagers. As none of the available randomised studies has been blinded or placebo controlled so far, and none has previously included young teenagers, our clinical trial is underpinned by a good medical and scientific rationale. The debate on the effect of ovarian protection using GnRHa or other means of gonadotropin suppression has lasted since the mechanism was hypothesised over 40 years ago[1] and our aim is to employ modern biochemical markers of ovarian reserve to investigate if the administration of GnRHa during cancer treatment can preserve the fertility of young female cancer subjects.

Here we report a summary of our protocol for a prospective phase III study, randomised, double-blinded, placebo-controlled, that adhere to International Council for Harmonisation-Good Clinical Practice (ICH-GCP)

and Standard Protocol Items: Recommendations for Interventional Trials (SPIRIT) guidelines. V.2.5 2022-09-27 of the protocol is available as online supplemental material 1 and any substantial amendments will be available on clinicaltrials.gov or by request from the corresponding author. The SPIRIT checklist refers to the full protocol.

## METHODS AND ANALYSES
### Hypothesis and aim

We hypothesise that if GnRHa has a protective effect on the ovarian reserve when given during chemotherapy, this effect will be reflected in effect measurements of biochemical markers of fertility such as AMH, FSH, LH, inhibin, estadiol and clinically measured outcomes such as antral follicle counts (AFC) on transvaginal ultrasonography and resumption of menses. Our primary objective is to estimate the changes in ovarian reserve markers following chemotherapy for treatment of cancer with or without GnRHa by determination of the AMH at 12 months after end of gonadotoxic treatment (EoT) in women with breast cancer. A full list of the study objectives are listed in table 2.

### Study design

A phase III, prospective, randomised, double-blinded, placebo-controlled study to investigate the effect of GnRHa as a protective agent during chemotherapy treatment. The study will include adult women and teenagers diagnosed with breast cancer, acute leukemias, lymphomas (Hodgkin and non-Hodgkin) or sarcomas (osteo, soft tissue and Ewing) treated with chemotherapy. The study will include approximately 300 subjects with breast cancer and up to 200 subjects with lymphoma, acute leukemias and sarcomas. After informed consent, baseline values will be collected and subjects will be randomised 1:1 to treatment with GnRHa (triptorelin) or placebo. Treatment will otherwise be according to best practice. The follow-up period starts at EoT and includes seven visits during the 5 year follow-up (figure 1).

### Setting, inclusion and study subjects

The study Sponsor is Karolinska University Hospital, Sweden and the study involves patient inclusion from 19 different sites around Sweden. Female subjects, aged 14–42 years at cancer diagnosis, with breast cancer or acute leukemias, lymphomas (Hodgkin and non-Hodgkin) or sarcomas (osteo, soft tissue and Ewing) as confirmed by histology and assigned for disease-specific chemotherapy will be included after signed informed consent. For study subjects under the age of 18, signed consent from both legal guardians are required in addition to consent from the study participant. Confirmed menarche, Eastern Cooperative Oncology Group (ECOG) performance status 0–1 and adequate bone marrow, renal, hepatic and cardiac functions and absence of other uncontrolled medical or psychiatric disorders are required for

inclusion. Full inclusion and exclusion criteria are available in the prerecorded protocol (online supplemental material 1). The study opened for inclusion Q2 2023 and has a planned 3 year inclusion period. Primary endpoint data are estimated to be ready for analyses Q4 2027 and end of study, defined as the last study subject's last follow-up visit, by Q4 2031.

### Randomisation

At the time of inclusion, inclusion and exclusion criteria are entered into a web-based randomisation application (ALEA) that will be operated by the Clinical Trials Office (CTO), Centre for Clinical Cancer Studies at Karolinska University Hospital. If all criteria are met, subjects are randomised to receive triptorelin or placebo in a 1:1 ratio using permuted block randomisation. Subjects will be stratified per diagnosis (breast cancer, lymphoma, sarcoma and leukaemia) and for subjects with breast cancer also for age groups (14–34; ≥35). The ALEA will assign a unique subject identification number and the randomisation arm to each subject. The subject's identification number will be used on all study-related documents including the electronic Case Report Form (eCRF).

### Blinding

All personnel involved in the study, sponsor, investigators, site staffs and monitors, statisticians and subjects, except personnel preparing triptorelin/placebo at the local site and an unblinded monitor, will be blinded during the study. An unblinded research nurse at each site will prepare blinded triptorelin/placebo to subjects and manage registration in the web-based randomisation system ALEA, used to allocate blinded triptorelin/placebo to subjects at randomisation.

### Study treatment

The study treatment includes the addition of GnRHa (triptorelin (Pamorelin), intramuscular injection) or placebo (intramuscular injection) in addition to the best practice disease-specific cytostatic treatment. The first dose of triptorelin/placebo is given between −14 and +7 days from the start of the disease-specific gonadotoxic treatment as an intramuscular injection.

One triptorelin (Pamorelin) injection holds a dose that lasts either 1 month (3.75 mg) or 3 months (11.25 mg). The number of doses per subject will be adjusted to cover the disease-specific gonadotoxic treatment with as few injections as possible. Most study treatments correspond to GnRHa/placebo treatment for a duration of 3–6 months (1–2 injections). The amount of triptorelin/placebo is calculated based on the length of each block of disease-specific cytostatic treatment. The end of disease-specific gonadotoxic treatment is also EoT, and the first follow-up is planned 6 months after EoT (figure 1). Further information on the included chemotherapeutic treatment regimens is available in the full study protocol (online supplemental material 1).

### Study interventions

At the screening/baseline visit, the principal investigator at each site will ensure that each subject will be provided with written and oral information about the study. The informed consent form is signed before any study-related activities are carried out. A physical examination and complete medical history including demographic data will be performed before start of study treatment. Information regarding concomitant medications including ongoing fertility preservation measures, use of contraceptives and hormonal treatments will be collected at screening/baseline visit and changes in concomitant medications will be assessed throughout the study. Any reported changes will be recorded in the eCRF. Blood samples for measurement of routine blood parameters (a complete blood count, including B-hemoglobulin, white blood cells, differential count and platelets, liver status (alkaline phosphatase), alanine transaminase, aspartate aminotransferase, billirubin and gamma-glutamyl transferase), renal status (sodium, potassium, calcium, creatinine and albumin)) will be collected at baseline and EoT and research blood samples (hormonal markers: AMH, FSH, inhibin, estradiol and LH) and a transvaginal ultrasound (AFC and ovarian blood flow) will be offered at each study visit.

Reproductive outcome data, reproductive health, sexual health and quality of life will be collected through validated questionaires including EORTC QLQ C30, Hospital Anxiety and Depression Scale (HAD) and Female Sexual Function Index (FSFI) .

A full list of study procedures and interventions at each study visit is available in table 3 and in the full protocol (online supplemental material 1).

### Study samples

The total volume of blood taken from each subject during the study is 200 mL; 40 mL (safety blood samples 2×20 mL) and 160 mL (research samples 8×20 mL) over a period from inclusion to 5 years from EoT. If analyses of study parameters are not immediate, research samples will be registered at Stockholms medicinska biobank at Karolinska University Hospital and handled according to the current biobank laws and regulations. The research samples will be coded/pseudonymised to protect the study subject's identification. Biobanked research samples will be analysed at Karolinska Institutet or Karolinska University Hospital, Sweden.

### Outcome measures and endpoints

Levels of the primary variable, AMH in women with breast cancer, will be measured at baseline, at EoT and continuously during follow-up 6 months, 12 months and years 2, 3, 4, 5 after EoT to establish the primary endpoint of difference in AMH recovery at follow-up 12 months after EoT, relative to AMH levels at EoT, and the secondary endpoint of difference in AMH recovery at follow-up until 5 years after EoT, relative to AMH levels at EoT, as compared between the GnRHa group and the placebo

**Table 3** Study visits and study procedures

| Procedure | Visit 1 Screening baseline | Visit 2 Day 0 (−14/±7 days) Initiation of study treatment (triptorelin/ placebo)/ gonadotoxic treatment | Treatment visits during gonadotoxic treatment (triptorelin/ placebo) (0–4 additional visits for study treatment) (±7 days) | Visit 3 EoT (±1 month) End of gonadotoxic treatment (1 or 3 months after last administration of triptorelin/placebo) | Visit 4 6 months (±1 months) From EoT | Visit 5–9 1,2,3,4,5 years (±1 months) From EoT |
|---|---|---|---|---|---|---|
| Inclusion/exclusion criteria | X | X | | | | |
| Informed consent | X | | | | | |
| Fertility preservation | X | | | | | |
| Demography | X | | | | | |
| Physical examination | X | | | | | |
| Vital signs | X | X | X | X | X | X |
| Medical history | X | | | | | |
| Randomisation | | X | | | | |
| Safety blood samples* | X | | | X | | |
| Pregnancy test (if older than 16 and sexually active) | X | | | | | |
| Biochemical markers† | X | | | X | X | X |
| Bone mineral density | X | | | X | | X‡ |
| Ultrasound§ | X | | | X | X | X |
| Study treatment¶ | | X | X | | | |
| Questionnaire (baseline) | X | | | | | |
| Questionnaire (follow-up) | | | | X | X | X |
| Concomitant medication | X | X | X | X | X | X |
| Adverse events (AE and SAE) | | X | X | X | X | X |

*Safety blood samples taken as clinical routine during the cancer treatment are recorded in the eCRF. Safety blood samples at baseline and EoT will be assessed as part of the study.
†Research samples including AMH, follicle stimulating hormone (FSH), inhibin, estradiol and LH.
‡Only at years 1 and 5 of follow-up.
§Antral follicle counts and ovarian Doppler flow.
¶Subjects randomised to treatment GnRHa will receive intramuscular injection(s) of triptorelin, 3.75 mg/month or 11.25 mg/3 months covering the duration of the gonadotoxic chemotherapy, subjects randomised to placebo will receive intramuscular injection(s) of placebo (NaCl 0.9%).
AE, adverse events; AMH, anti-Müllerian hormone; eCRF, electronic Case Report Form; EoT, end of treatment; GnRHa, gonadotropin-releasing hormone agonists ; LH, luteinising hormone; SAE, serious adverse events.

group in women with breast cancer. The timing of the primary and secondary outcome analyses were based on longitudinal AMH measurements postchemotheraphy available from previous studies.[18 26]

Levels of the secondary variables, AMH in women with lymphomas, sarcomas and leukemias, FSH, inhibin and estradiol, AFC and ovarian blood flow, menstruation data and quality of life parameters will be measured at baseline, at EoT and continuously during follow-up 6, 12, months and years 2, 3, 4, 5 after EoT. Markers of bone mineral density will be measured at baseline and EoT, 1 and 5 years after EoT. A full list of secondary endpoints

are in the study protocol (online supplemental material 1).

## End of study

The end of study is defined as the last study subject's last follow-up visit. The principal investigator has the right to terminate the study for clinical or administrative reasons at any time. The study may be prematurely terminated due to a high number of serious adverse events (related or not related to the study medication) or if the enrolment process cannot be completed within a reasonable time frame.

## Data management and monitoring

The requirements regarding information in the medical records adhere to the Patient Data Act (SFS 2008:355) and 'The Medical Product Agency's regulations on clinical trials of medicinal products for human use' (LVFS 2011:19). The investigator must keep source documents for each subject in the study. An eCRF is used for data collection. The eCRF is provided in the web-based SAS system PheedIt. This database is provided by Region Stockholm and is registered at CTO, Centre for Clinical Cancer Studies, Cancer Theme, Karolinska University Hospital, Stockholm, Sweden. For details and information that is study specific and of no interest for the medical care of the subject, CRF and other documents may be considered as source data.

Prior to study start the expected source location of source data (eg, medical record, laboratory reports, validated questionnaires and the eCRF), must be identified and documented. This will be done by completing a site-specific source data list. This list defines what documents contain the source data for each study specific parameter. All information processed by the sponsor will be pseudonymized (coded) and study subjects will be identified by a study ID.

A monitor from the CTO at Centre for Clinical Cancer Studies (CKC), Cancer Theme, Karolinska University Hospital will be responsible for coordinating the monitoring activities of the study and ensure adherence to ICH-GCP guidelines. The monitor will have access to medical records and source data after a secrecy agreement has been signed by the responsible party at the site as well as by the monitor. The investigator must ensure that all source documents are accessible for monitoring and other quality control activities.

## Sample size calculation and statistical analyses

The sample size calculation was done assuming a mean difference in log10-transformed AMH-levels of 0.22 between GnRHa group and placebo group (with a ratio 1:1). It was further assumed that data are approximately normally distributed, and that each treatment group has a SD of 0.63.[26] Under these assumptions, the hypothesis test (a two-sided test with significance level 5%) of the mean effect between GnRHa group versus placebo group will have a power of at least 80% if the sample size is 300 (including an assumed 15% drop-out rate).

Primary and secondary endpoints will be evaluated using the intention-to-treat (ITT) (ie, all randomised subjects), the per protocol (PP) analysis set (ie, all randomised subjects with no major protocol violations) and the safety analysis set (all subjects) (figure 2). The classification of each subject with respect to each analysis set will be done prior to database lock.

Continuous variables will be summarised as number of subjects (n), mean, median, SD, quartile 1 (Q1) and quartile 3 (Q3) and range (min, max) by visit. The

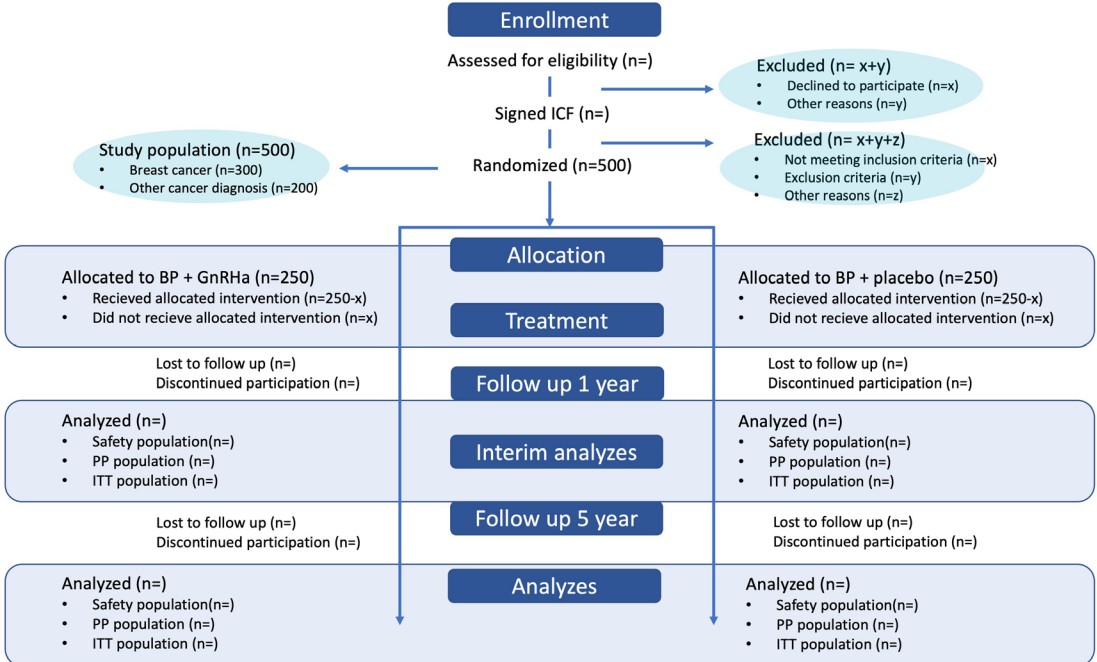

**Figure 2** Study flow chart. GnRHa, gonadotropin-releasing hormone agonists; ITT, intention to treat; PP, per protocol.

change from baseline at each respective visit will also be presented. Discrete (categorical/ordinal) variables will be summarised in frequency tables (frequency and proportion) by visit. All statistical tests will be two-sided assuming a 5% significance level.

Statistical analyses will be performed using SAS (V.9.4 or higher, SAS Institute, Cary, NC, USA) unless otherwise stated.

## Data analysis plan

An interim report will be produced after last enrolled subject has completed follow-up 12 months after EoT (primary endpoint). The interim report will include analyses of the primary and some secondary endpoints based on data assembled for each enrolled subject up until the time point corresponding to 12 months after EoT. The assembled data will be locked and the study unblinded only for the statisticians. Main analysis for the primary endpoint will be done using the ITT population. Analysis on the PP population will be performed as sensitivity analysis. Subgroup analysis will be performed on ITT population for AMH change on these prospectively defined subgroups:

► Age (14–19; 20–34; ≥35)
► Ovarian reserve (AMH) at baseline (0.68–2.3; 2.4–4.9; ≥5 (µg/L)).
► Cancer diagnosis (breast cancer, lymphoma, sarcoma and leukaemia)
► Type of cancer treatment (high risk, intermediate risk and low risk).
► Exposure to gonadotoxic cytostatics (corresponding to cumulative dose, mg CPA)/.

Analyses results and aggregated descriptive summaries will be disclosed to the sponsor.

At the end of study, after the extended follow-up period of 5 years, and once all study data for the mentioned visits are collected and verified in the eCRF system, the unblinding of all subjects will take place for investigators and sponsor. All available data will be used in the statistical analysis. A subject who withdraws prior to the last planned visit in the study will be included in the analyses up to the time of discontinuation. All efficacy and safety analyses will be performed at end of study when the study database is released for unblinding.

## Safety and adverse events

The study drug is well established for other indications in Sweden and used off-label for this indication in Sweden and globally. During the study, we will register standard safety blood samples at baseline and at EoT as well as access the safety blood markers taken during chemotherapy through the subject's medical records. At each study visit, adverse events (AE) related to the study product are registered, starting at study randomisation and up to 90 days after the last dose of the investigational products, triptorelin and placebo (sodium chloride). All AE that occur during the study and which are observed by the investigator/study nurse or reported by the subject will be registered in the eCRF with complete information regarding seriousness (mild, moderate and severe), intensity and causality. Serious adverse events (SAE) will be reported by the investigator to the sponsor on a separate SAE form within 24 hours. Those SAE which are assessed by sponsor to be Suspected Unexpected Serious Adverse Reactions are further reported via a CIOMS form to the European Medicines Agency (EudraVigilance database) within their specified time frames.

While the study is ongoing, a Development Safety Update Report (DSUR) will be reported annually to the MPA by the sponsor. A monitor from the CTO at CKC, Cancer Theme, Karolinska University Hospital will be responsible for coordinating the monitoring activities of the study and ensure adherence to ICH-GCP guidelines. Authorised representatives for the sponsor and Competent Authorities may carry out audits or inspections at the study site, including source data verification. The investigator must ensure that all source documents are available for audits and inspections.

## Patient and public involvement

Patients, patient relatives and patient associations took part in the study design. A patient representative from the Swedish network for Young Women with Breast Cancer acts as a designated consultant for this study. Patient involvement will be continued throughout the study, with the aim to informing on, discuss and improve all aspects of the project, according to the framework suggested by INVOLVE (http://www.invo.org.uk).

## ETHICS, PERMITS AND DISSEMINATION

This study is designed and shall be implemented, executed and reported in accordance with the study protocol, principles of the International Conference on Harmonisation Tripartite Guideline for Good Clinical Practice (ICH-GCP E6 (R2)), applicable local regulations (ie, European Directive 2001/20/EC) and the ethical principles of the Guidelines of the World Medical Association (WMA) Declaration of Helsinki (as amended by the 64th WMA General Assembly, Fortaleza, Brazil, October 2013). The National Institutional review board in Sweden (Etikprövningsmyndigheten, dnr: 2021-03379, approval date 12 October 2021 (approved amendments 12 June 2022, dnr: 2022-02924-02 and 2022-12-13, dnr: 2022-05565-02). Approval by the Swedish Medical Product Agency (Läkemedelsverket) 19 January 2022, Dnr: 5.1-2021-98927 (approved amendment 4 February 2022). Manufacturing authorisation for authorised medicinal products approved 6 December 2021, Dnr: 6.2.1-2020-079580. Biobankpermit from Stockholm Medical Biobank (SMB) approved 22 June 2022, RBC dnr: 202 253. No substantial changes to the study protocol can be made without renewed approval from the Swedish Ethical Review Authority or the Swedish Medical Product Agency. Changes affecting the study subjects will be reflected in the informed consent form.

Within 6 months of study completion (due to study involving teenagers), the results shall be analysed, a clinical study report with individual data shall be prepared and the study results shall also be reported in the EudraCT database. Study results will be published in open access peer reviewed scientific journals.

**Author affiliations**
¹Department of Oncology-Pathology, Karolinska Institute, Stockholm, Sweden
²Department of Reproductive Medicine, Karolinska University Hospital, Stockholm, Sweden
³Theme cancer, Karolinska Comprehensive Cancer Center and University Hospital, Stockholm, Sweden
⁴Pediatric Theme Astrid Lindgren's Pediatric Hospital, Stockholm, Sweden
⁵Department of Cellular Therapy and Allogeneic Stem Cell Transplantation, Karolinska University Hospital, Stockholm, Sweden
⁶Division of Hematology, Department of Medicine Huddinge, Karolinska Institute, Huddinge, Sweden
⁷Department of Oncology, Capio ST, Göran Hospital, Stockholm, Sweden
⁸Department of Oncology, Sahlgrenska University Hospital, Goteborg, Sweden
⁹Oncology, Örebro universitet Fakulteten för medicin och hälsa, Orebro, Sweden
¹⁰Department of Oncology, Norrlands University Hospital, Umeå, Sweden
¹¹Department of Hematology, Cancer Theme, Karolinska University Hospital, Stockholm, Sweden
¹²Department of Hematology and Coagulation, Sahlgrenska University Hospital, Goteborg, Sweden
¹³Akademiska Hospital, Uppsala, Sweden
¹⁴Center for Pediatric Oncology, Skåne University Hospital, Lund, Sweden
¹⁵Department of Medicine, Karolinska Institute, Solna, Sweden
¹⁶Division of Clinical Epidemiology, Department of Medicine Solna, Karolinska Institute, Stockholm, Sweden

**Contributors** Conceptualisation: KAR-W, SE, JB, PL, JM, BL, AV and HM. Funding acquisition: KAR-W. Methodology: KAR-W, SE, HPN, JM, AV and HM. Project administration: HPN and KAR-W. Scientific contribution: KAR-W, HPN, SE, JB, PL, JM, BL, AV, HM, TF, CLS, EIF, AA, SH, LV and PF. Resources: KAR-W, HPN, SE, JB, PL, JM, BL, AV, HM, TF, CLS, EIF, AA, SH, LV and PF. Method validation: KAR-W, HPN, SE, JB, PL, JM, BL, AV, HM, TF, CLS, EIF, AA, SH, LV and PF. Writing and revising the protocol: KAR-W, HPN, SE, JB, PL, JM, BL, AV, HM, TF, CLS, EIF, AA, SH, LV and PF. KAR-W is study sponsor and responsible for future analyses and publication of study outcomes. All authors have read and agreed upon the final manuscript and approved the submitted version.

**Funding** This work was supported by Vetenskapsrådet (KBF 2019–00446), the Swedish Cancer Society (190249Pj, 20 0170F), Radiumhemmets Research Funds (201313, N/A) and the Swedish Childhood Cancer Foundation (KP2022-0013).

**Competing interests** None declared.

**Patient and public involvement** Patients and/or the public were involved in the design, or conduct, or reporting or dissemination plans of this research. Refer to the Methods section for further details.

**Patient consent for publication** Not required.

**Provenance and peer review** Not commissioned; externally peer reviewed.

**ORCID iDs**
Kenny A Rodriguez-Wallberg http://orcid.org/0000-0003-4378-6181
Hanna Pauline Nilsson http://orcid.org/0009-0009-8218-3835
Antonis Valachis http://orcid.org/0000-0001-6059-0194
Anne Andersson http://orcid.org/0000-0002-9597-6465

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
