## [Reviewer comments · BMJ Open]

ARTICLE DETAILS

TITLE (PROVISIONAL)	ProFertil Study Protocol for the investigation of Gonadotropin Releasing Hormone agonists (GnRH α) during Chemotherapy aiming at Fertility Protection of Young Women and Teenagers with Cancer in Sweden – A Phase III Randomized Double-Blinded Placebo-Controlled Study
AUTHORS	Rodriguez-Wallberg, Kenny; Nilsson, Hanna; Bergh, Jonas; Malmros, Johan; Ljungman, P; Foukakis, Theodoros; Stragliotto, Christina Linder; Friman, Erika Isaksson; Linderholm, Barbro; Valachis, Antonis; Andersson, Anne; Harrysson, Sara; Vennström, Lovisa; Frisk, Per; Mörse, Helena; Eloranta, Sandra

VERSION 1 – REVIEW

REVIEWER	Marin, Loris University of Padua
REVIEW RETURNED	23-Aug-2023

GENERAL COMMENTS	I congratulate the authors for designing this phase III, randomized, double-blind, placebo-controlled study of the use of GnRH α during chemotherapy for the fertility protection of young women and adolescents with cancer. The efficacy of GnRH administration during chemotherapy in preserving fertility and ovarian function has long been debated and this study may finally lead to an answer. The study design and methodology are well presented and the research appears ethical.
---

REVIEWER	Odendaal, Joshua University of Warwick
REVIEW RETURNED	25-Aug-2023

GENERAL COMMENTS	This is a protocol paper for an RCT assessing the role of GnRHα during chemotherapy in women undergoing chemotherapy for cancer of various primaries. Overall the study addresses an important research question in an area that needs further high quality trials. Several comments as below: I am unclear at what stage the study is currently at? It would be worth inserting a paragraph outlining whether recruitment has commenced or when it is expected to and when follow-up is anticipated to end. Line 117-125: Whilst I agree with the authors that the majority of existing studies are uncontrolled it should be acknowledged that some RCTs in the area do exist as an example: doi:10.7150/jca.31859 or doi.org/10.1093/annonc/mdx184
--

	Line 202-204: Given that study treatments may be in two different dosages dependent on chemotherapy regime will this form a minimisation criteria? Presumably this is not the case as not already stated but please state if so. Given the length of follow-up it is likely a subset will require further courses of chemotherapy, how will these be dealt with within the trial framework? The following typographical errors need correcting: Line 114: Should read Expenses Line 314: Should read Intermediate
--	--

REVIEWER	Stockler, Martin University of Sydney , NHMRC Clinical Trials Centre
REVIEW RETURNED	30-Sep-2023

GENERAL COMMENTS	I enjoyed reading this well-conceived and beautifully written protocol paper for a placebo controlled RCT addressing an important, practical, and unresolved issue for young females being treated with chemotherapy. I agree with the rationale for the trial, but the explanation of possible mechanisms in the introduction, lines 96-105, is hard to understand, and should be revised to improve clarity. Lines 105-115 are fine. The primary analysis seems to be a comparison of mean log-AMH levels between the randomly assigned groups. However, in the outcome measures and endpoints section, lines 243-245, it is stated that the primary endpoint is “recovery of AMH levels at follow-up 12 months after EoT”; and secondary endpoints are “recovery of AMH levels at follow up until 5 years after EoT, as compared between the GnRHa group and the placebo group in women with breast cancer.” This implies a binary outcome of AMH recovered, yes versus no”. The proposed primary analysis comparing mean log-AMH levels between groups is fine, and will provide higher power than a comparison based on a binary outcome. However, it would be useful to also provide a precise, quantitative definition of ‘recovered AMH level’, that could be used as in a supplementary analysis that compares the numbers (proportions) of participants in each group with a ‘recovered AMH level’, as a more patient-centred endpoint.
--

VERSION 1 – AUTHOR RESPONSE

Reviewer: 1
Dr. Loris Marin, University of Padua
Comments to the Author:

 I congratulate the authors for designing this phase III, randomized, double-blind, placebo-controlled study of the use of GnRHa during chemotherapy for the fertility protection of young women and adolescents with cancer. The efficacy of GnRH administration during chemotherapy in preserving fertility and ovarian function has long been debated and this study may finally lead to an answer. The study design and methodology are well presented and the research appears ethical.

- Thank you for your comments.

Reviewer: 2

Dr. Joshua Odendaal, University of Warwick, University Hospitals Coventry and Warwickshire NHS Trust

Comments to the Author:

This is a protocol paper for an RCT assessing the role of GnRHa during chemotherapy in women undergoing chemotherapy for cancer of various primaries. Overall the study addresses an important research question in an area that needs further high quality trials. Several comments as below:

I am unclear at what stage the study is currently at? It would be worth inserting a paragraph outlining whether recruitment has commenced or when it is expected to and when follow-up is anticipated to end.

- Thank you for your comments. The study opened for inclusion Q2 2023 and we have added this, as well as information on planned End of Study to the methodology section.

 Line 117-125: Whilst I agree with the authors that the majority of existing studies are uncontrolled it should be acknowledged that some RCTs in the area do exist as an example:
doi:10.7150/jca.31859 or doi.org/10.1093/annonc/mdx184

- Yes, while we do include a full table on the concluded RCTs on GnRHa (Table 1), we should perhaps have clarified in the text what the table refers to as we understand that the current paragraph might read as though we do not at all acknowledge the existing studies. Thank you for observing this.

 Line 202-204: Given that study treatments may be in two different dosages dependent on chemotherapy regime will this form a minimisation criteria? Presumably this is not the case as not already stated but please state if so.

- Thank you for your comment. All women will receive the same dosage, but the Triptorelin to be used in this study comes with two different presentations allowing one injection effective for one month, and a long-term injection with effect for 3 months.

 Given the length of follow-up it is likely a subset will require further courses of chemotherapy, how will these be dealt with within the trial framework?

- Thank you for a very interesting question. While most cases of breast cancer recurrence occur within 5 years, we consider it unlikely that further treatment will significantly affect the 1 year follow up based on the approximate 90% five-year survival in breast cancer patients. Power was calculated based on a drop-out rate of 15% where this possibility was considered. Any patients in need of further gonadotoxic treatment during follow-up will still be part of the study but excluded from the Per Protocol population from that point on.

 The following typographical errors need correcting:

Line 114: Should read Expenses

Line 314: Should read Intermediate

- Thank you, this is corrected!

Reviewer: 3

Prof. Martin Stockler, University of Sydney

Comments to the Author:

 I enjoyed reading this well-conceived and beautifully written protocol paper for a placebo controlled RCT addressing an important, practical, and unresolved issue for young females being treated with chemotherapy. I agree with the rationale for the trial, but the explanation of possible mechanisms in the introduction, lines 96-105, is hard to understand, and should be revised to improve clarity. Lines 105-115 are fine.

- Thank you for your considerate comments and kind suggestions. Whilst it is challenging to provide an mechanistic explanation for an effect that is yet unknown, we have attempted to clarify our reasoning about this in lines 96-105 . This section now reads:

“The primary mechanism proposed for GnRHa-mediated protection of the ovarian reserve involves keeping the primordial follicles dormant throughout chemotherapy by antagonistic occupation of the GnRH receptor, leading to desensitization of the pituitary gland and suppression of follicle stimulating hormone (FSH) and luteinizing hormone (LH) secretion, with a profound hypogonadal effect achieved through receptor downregulation by internalization of receptors. But primordial follicles are non-proliferating, and not regulated by gonadotropins as they lack both FSH and GnRH receptors.² Thus, administration of GnRHa should not be able to block their initial recruitment towards maturation, at least not through the proposed mechanism.^{3,4,5} If GnRHa has an antagonistic effect on the ovarian cytotoxicity of chemotherapeutics, as has been suggested, systemic use of GnRHa would likely reduce the overall effect of chemotherapy, which does not seem to be the case.⁵ Nevertheless, it is possible that prophylactic GnRHa therapy acts through mechanisms yet unknown.”

 The primary analysis seems to be a comparison of mean log-AMH levels between the randomly assigned groups. However, in the outcome measures and endpoints section, lines 243-245, it is stated that the primary endpoint is “recovery of AMH levels at follow-up 12 months after EoT”; and secondary endpoints are “recovery of AMH levels at follow up until 5 years after EoT, as compared between the GnRHa group and the placebo group in women with breast cancer.” This implies a binary outcome of AMH recovered, yes versus no”.

The proposed primary analysis comparing mean log-AMH levels between groups is fine, and will provide higher power than a comparison based on a binary outcome. However, it would be useful to also provide a precise, quantitative definition of ‘recovered AMH level’, that could be used as in a supplementary analysis that compares the numbers (proportions) of participants in each group with a ‘recovered AMH level’, as a more patient-centred endpoint.

- We are sorry to hear that our outcome unintentionally reads as a binary approach. To clarify that we compare levels of recovery we now state “difference in recovery of AMH levels at follow-up 12 months after EoT” as is indeed the original sentence from the full study protocol.

While it is a possible option to investigate clinical recovery defined as post treatment AMH levels above clinical values for ovarian insufficiency, we don’t believe adding this as a binary outcome parameter will be of much value in a heterogenous cohort where a large proportion of the study participants will naturally be closing in on the end of their reproductive lifespan.

VERSION 2 – REVIEW

REVIEWER	Odendaal, Joshua University of Warwick
REVIEW RETURNED	30-Oct-2023

GENERAL COMMENTS	Overall I am happy that the authors have addressed my previous comments and wish them well in their study.
REVIEWER	Stockler, Martin University of Sydney , NHMRC Clinical Trials Centre
REVIEW RETURNED	31-Oct-2023
GENERAL COMMENTS	Suggestions addressed.